# Ploidy Levels and Fitness-Related Traits in Purebreds and Hybrids Originating from Sterlet (*Acipenser ruthenus*) and Unusual Ploidy Levels of Siberian Sturgeon (*A. baerii*)

**DOI:** 10.3390/genes11101164

**Published:** 2020-10-02

**Authors:** Sahana Shivaramu, Ievgen Lebeda, Doi Thi Vuong, Marek Rodina, David Gela, Martin Flajšhans

**Affiliations:** South Bohemian Research Center for Aquaculture and Biodiversity of Hydrocenoses, Faculty of Fisheries and Protection of Waters, University of South Bohemia in České Budějovice, Zátiší 728/II, 389 25 Vodňany, Czech Republic; ilebeda@frov.jcu.cz (I.L.); doivuong@gmail.com (D.T.V.); rodina@frov.jcu.cz (M.R.); gela@frov.jcu.cz (D.G.); flajsh@frov.jcu.cz (M.F.)

**Keywords:** sturgeons, polyploidy, aquaculture, restocking, fitness-related traits

## Abstract

The present study aimed to investigate and compare fitness-related traits and ploidy levels of purebreds and hybrids produced from sturgeon broodstock with both normal and abnormal ploidy levels. We used diploid *Acipenser ruthenus* and tetraploid *A. baerii* males and females to produce purebreds and reciprocal hybrids of normal ploidy levels. Likewise, we used diploid *A. ruthenus* and tetraploid *A. baerii* females mated to pentaploid and hexaploid *A. baerii* males to produce hybrids of abnormal ploidy levels. Fertilization of ova of *A. ruthenus* and *A. baerii* of normal ploidy with the sperm of pentaploid and hexaploid *A. baerii* produced fully viable progeny with ploidy levels that were intermediate between those of the parents as was also found in crosses of purebreds and reciprocal hybrids of normal ploidy levels. The *A. ruthenus* × pentaploid *A. baerii* and *A. ruthenus* × hexaploid *A. baerii* hybrids did not survive after 22 days post-hatch (dph). Mean body weight and cumulative survival were periodically checked at seven-time intervals. The recorded values of mean body weight were significantly higher in *A. baerii* × pentaploid *A. baerii* hybrids than other groups at three sampling points (160, 252 and 330 dph). In contrast, the highest cumulative survival was observed in *A. baerii* × *A. ruthenus* hybrids at all sampling points (14.47 ± 5.70 at 497 dph). Overall, most of the studied sturgeon hybrids displayed higher mean BW and cumulative survival compared to the purebreds. The utilization of sturgeon hybrids should be restricted to aquaculture purposes because they can pose a significant genetic threat to native populations through ecological interactions.

## 1. Introduction

Sturgeons (Acipenseridae) are one of the most unique and oldest families of fishes that are often considered to be “living fossils” because the fossil record of sturgeons dates back to Upper Cretaceous [1,2]. Evidence suggests that the sturgeon lineage diverged from the lineage that gave rise to ancient pre-Jurassic teleosts approximately 300 million years ago (MYA) according to their mitochondrial DNA analysis [2]. Sturgeons are one of the most endangered fishes in the world [3] because the sturgeon populations have experienced a dramatic decline throughout their range due to overfishing, habitat destruction and other anthropogenic causes [3,4,5]. There are 27 species of sturgeons in the world that are found along the coast of the Pacific and Atlantic oceans and in seas, inland lakes and rivers in the northern hemisphere [4]. The family, Acipenseridae, includes the following four genera: Acipenser, Huso, Scaphirhynchus, and Pseudoscaphirhynchus [4].

Sturgeons are well known for their polyploid origin with at least three independent genome duplication events taking place during their evolution, and more genome duplication events are expected [6]. Several evolutionary structures in sturgeons have been proposed by the different groups of scientists to explain the origin of their polyploidy with theories on lineage-specific multiple genome duplications [7,8] to that of a combination of auto and allopolyploidy [9] and numerous hybridizations leading to allopolyploidization [10,11]. Approximately two hundred million years of evolution and various ploidy levels make sturgeons a distinct category for studying the evolution of polyploidy in fishes. Most recently, researchers discovered that the whole genome duplication process included the loss of a whole chromosome in sterlet (*A. ruthenus*). They also found that the decrease in redundancy of the polyploid genome is highly random and the functional tetraploidy in sterlet is mainly conserved by adaptive processes over 180 million years [12]. 

Currently, there are three well-classified groups of acipenseriform species depending on the number of chromosomes and DNA content in their cell nuclei. The three distinct groups include species with ~120 (*A. ruthenus*), ~240 (*A. baerii*) and ~360 (*A. brevirostrum)* chromosomes, with respect to their increased DNA content [6]. Havelka et al. [13] pointed out that their ploidy levels are distinguished either on an evolutionary scale that refers to ancient ploidy levels and assumes tetraploid (4n)–octaploid (8n)–dodecaploid–(12n) relationships [7], or on a functional scale that originates from significant functional genome rediploidization during the evolution of sturgeon and assumes diploid (2n)–tetraploid (4n)–hexaploid (6n) relationships [6,8,9]. For clarity, all ploidy levels in this study refer to the functional scale.

Hybridization of the sturgeon species with different ploidy levels is reported both in nature [14,15,16,17] and captivity [18] probably since their genomes exhibit a polyploid nature. While natural hybridization of sturgeon species is supposed to be a severe threat for their endangered populations [16], artificially intended hybridization is often utilized in aquaculture for attaining the vigor of the hybrids [18]. The hybridization of sturgeons with different ploidy levels can give rise to offspring with abnormal ploidy levels. Sturgeons with abnormal ploidy levels have already been reported in the natural waters [16] and in aquaculture facilities [13,19,20,21,22,23]. The most likely reason for the origin of these individuals with abnormal ploidy levels is due to autopolyploidization processes such as the spontaneous diploidization of the maternal chromosome set (SDM) evidenced in sturgeons [13,23] or polyspermic fertilization [24] or interspecific hybridization [16,17].

Generally, interspecific hybrids of distantly related parental species with a different chromosome number are usually sterile because their chromosomes cannot pair correctly during the zygotene stage of meiosis prophase I and such kind of impairment interferes with gonadal development and gametogenesis [25]. It was thus assumed that hybrids of two species with different parental ploidy levels are sterile. Recent observation on the gamete development of the sturgeon hybrids originating from mating parents of varying ploidy levels revealed the possibility of limited fertility in the F1 hybrid males which is a significant breakthrough in sturgeon gonadal development [26,27]. Overall, it implies the common speculation of sterility of the hybrid sturgeon with abnormal ploidy can be fallacious and more importantly, the ploidy status and fitness characteristics of the produced F1 generation can be seriously questionable. While the importance of hybridization in the sturgeon breeding has been substantially studied [18,28,29,30], extensive research has not been conducted yet on the influence of polyploidy in cultured sturgeon. Today, sturgeon farming is a rapidly growing branch of aquaculture for the production of black caviar and boneless meat [31]. To meet the global market demand of black caviar, the commercial sturgeon farms are extensively using cultured broodstock that necessitates the need to understand the physiological fitness in polyploid hybrids. To date, no studies have assessed the fitness fate of the F1 hybrids produced from mating the sturgeon species of different ploidy levels. However, the reproducibility of different sturgeon species with abnormal ploidy levels is reported for aquaculture purposes [22,32,33]. Therefore, it is interesting to evaluate the fitness-related traits of the F1 hybrids obtained by crossing sturgeon species with abnormal ploidy levels. 

In the present study, we evaluated the reproductive features (fertilization rate and hatching rate), fitness-related traits such as body weight (BW), cumulative survival, growth heterogeneity (GH) and specific growth rate (SGR) of the first-generation purebreds and hybrids produced by crossing individuals of sterlet (*A. ruthenus*) and Siberian sturgeon (*A. baerii*) with normal and abnormal ploidy levels. We also analyzed the ploidy levels of the parents, purebred and hybrid groups using flow cytometry. Currently, Siberian sturgeon is the most commonly cultured sturgeon species in the world; hence we believe that this study will provide an essential insight into the global sturgeon aquaculture industry [28]. 

## 2. Materials and Methods

### 2.1. Origin, Maintenance and Artificial Propagation of Broodstock

Sturgeon broodstock was obtained from the Genetic Fisheries Center (GFC) at the Faculty of Fisheries and Protection of Waters, University of South Bohemia in České Budějovice, Czech Republic. One spontaneous pentaploid *A. baerii* male after cytometric examination; three males of *A. ruthenus* and three males of *A. baerii* were kept along with four females of *A. baerii* and six females of *A. ruthenus*. Fish were acclimatized in 4 m^3^ hatchery tanks maintained with a water flow rate of 0.2 L/s, O_2_ of 7.0 mg/L and water temperature of 15 °C for seven days (Table 1). The final maturation was artificially induced by hormonal treatment in both males and females. The ovulated eggs were collected after micro-incision of the oviduct according to protocols from Štěch et al. [34]. The intramuscular injection of the carp pituitary powder dissolved in 0.9% (*w/v*) NaCl solution at 4 mg/kg body weight induced the spermiation in males artificially. After 24 h, sperm were collected from the urogenital tract using a catheter of 5 mm diameter connected to a 20 mL plastic syringe and later it was transferred into a 100 mL tissue culture flask [35]. Sperm was stored at 4 °C until the spermatozoon motility was evaluated by microscopy [36] and sperm from the males with the highest spermatozoa motility was used for further activation and fertilization. A cryopreserved sperm sample (year-old cryopreserved sample that was collected from the male from the previous year’s artificial propagation) from one 15-year-old spontaneous hexaploid *A. baerii* male was also used in the experiment. Thawing of the sperm sample was performed in a water bath at 40 °C for 5 s, assessed for motility and was further used for fertilization [36].

### 2.2. Fertilization and Hatching

By using a factorial mating design, eight crosses were produced, in which two were purebred crosses: *A. ruthenus* (St♀ × St♂) and *A. baerii* (S♀ × S♂), while the other six were hybrid crosses: *A. ruthenus* × *A. baerii* (St♀ × S♂), *A. baerii* × *A. ruthenus* (S♀ × St♂), *A. ruthenus* × pentaploid *A. baerii* (St♀ × S♂ 5n), *A. ruthenus* × hexaploid *A. baerii* (St♀ × S♂ 6n), *A. baerii* × pentaploid *A. baerii* (S♀ × S♂ 5n) and *A. baerii* × hexaploid *A. baerii* (S♀ × S♂ 6n). To produce each cross, an equal number of eggs from the females were pooled together (80 g of eggs were collected from six females of *A. ruthenus* and 120 g of eggs was collected from four females of *A. baerii* individually) and divided into plastic beakers in 40 g aliquots according to the number of males. The plastic beakers containing aliquots of eggs were then placed on an electronic shaker just before the fertilization which was set at a speed of 200 rpm and 10 mm deflection. To obtain equal genetic contribution from individual males and to avoid sperm competition between different males, each aliquot was inseminated with 1 mL sperm from one of the three males per cross, except for the pentaploid and hexaploid Siberian sturgeon and sperm were immediately activated by pouring in 200 mL dechlorinated water for fertilization. All the aliquots per cross were pooled together into a circular bowl after fertilization, and a clay suspension of 20 g L^−1^ was added three minutes after fertilization to remove egg stickiness and left on the electronic shaker for 50 min. Later on, the pooled aliquots were repeatedly washed with water to remove the clay remnants and then incubated in glass jar incubators in triplicates [35]. The UV sterilized re-circulating water with an O_2_ saturation of 9 mg L^−1^ and water temperature of 15.0 °C was continuously supplied to the glass jars during the incubation period. Around 100 eggs were randomly sampled from each cross 6 h post-fertilization, in triplicate, to estimate the fertilization rate. The live embryos were counted at the 2nd or 3rd cleavage division, as previously explained [37]. The egg clumps and dead eggs were removed by siphoning the water regularly. The larvae that hatched out in 5–7 days were regularly separated from the dead larvae and shells in order to reduce the incidence of the fungal infection and consequent larval mortalities.

### 2.3. Fish Rearing

The larvae of each cross were initially reared in a 0.3 m^3^ separate indoor trough system after complete hatching. The larvae were not fed on any natural or artificial feed until the complete yolk sac absorption. The larvae were firstly fed with diced sludge worms (*Tubifex tubifex*) ad libitum for the initial two weeks. They were then shifted to co-feeding with a combination of diced sludge worms and formulated dry feed. The larval feeding completely turned to the dry formulated feed after four weeks of co-feeding. The various formulated feeds fed to the fishes throughout the experiment were from Alltech Coppens, the Netherlands. The larvae of each cross moved to separate 3.5 m^3^ indoor tanks, which were maintained with the average temperature of 22 °C for separate group-nursing after 100 days of initial rearing. The larvae of each cross were separately stocked with an initial stocking density of 7 kg/m^3^, and they were fed ad libitum with a formulated commercial feed (Coppens^®^ Start Premium; Coppens International B.V., Helmond, The Netherlands) containing 54% protein, 15% fat, 1% crude fiber and 9.4% ash. The 160 dph juveniles were color-marked with subcutaneous Visible Implant Elastomers (Northwest Marine Technologies Inc., Anacortes, WA, USA) on the inward ventral side of the rostrum to indicate the origin of the fish. These color-marked fishes were periodically checked, and an equal number from each group were stocked for communal rearing in triplicates with uniform environmental conditions, such as continuous partial water exchange, aeration rate, feeding rate and photoperiod. The Individual Passive Integrated Transponder (PIT) tags (134.2 kHz; AEG Company, Ulm, Germany) were subcutaneously implanted into the fish groups on 330 dph. The juvenile fishes were transferred to 4 m^3^ indoor circular tanks at 4 °C without feeding for overwintering at the stocking density of ~7 kg m^3^ after the second summer. After wintering, the fish were transferred to the 3.5 m^3^ outdoor circular tanks, which were maintained in an average temperature of 22 °C and were daily fed on a commercial diet of 4% of total fish biomass (Coppens^®^ Supreme-10 containing 49% of protein, 10% of fat, 0.8% of crude fiber and 7.9% of ash).

### 2.4. Measurement of Fitness-Related Traits and Mid-Parental Heterosis 

Fishes were periodically measured to calculate the cumulative survival rate and mean BW on 22, 65, 160, 252, 330, 424, and 497 dph. The specific growth rate (% day^−1^) was calculated as SGR = 100 × (lnW_2_ − lnW_1_) × ∆T^−1^, where W_1_ and W_2_ are the initial and final mean body weight, and ∆T is the time interval between sampling measurements in days. Likewise, the growth heterogeneity was calculated from CV_FBW_/CV_IBW_, in which CV is the coefficient of variation (100 × SD/mean) and IBW and FBW are the initial and final mean body weight.

The mid-parental heterosis for the cumulative survival and mean BW and of hybrid crosses were calculated as mid-parental heterosis = ((F1 − MP)/MP) × 100, in which F1 = the value of F1, and MP = the mean value of the two parents (purebred groups).

### 2.5. Flow Cytometry and DNA Content Analysis

The ploidy analysis of the broodstock was conducted using the sample of peripheral blood that was collected into a heparinized syringe from the caudal vessel of the fish. The blood sample was mixed with physiological saline (two drops blood 1 mL^−1^ saline), kept at 4 °C and after that treated with a 1 mL CyStain DNA 1 step staining solution (Sysmex Europe GmbH, Norderstedt, Germany). The blood samples were investigated by flow cytometry (Partec CCA I; Partec GmbH, Münster, Germany) in order to measure the relative DNA content per cell. The calibration of the device was undertaken using ready-to-use, DNA-stained and fixed rainbow trout erythrocytes. The erythrocytes of a functionally diploid *A. ruthenus* with a relative DNA content of 3.74 pg DNA per cell [14] was used as the reference diploid standard and its spermatozoa was used as the reference haploid standard for comparison of the different ploidy levels.

The ploidy level of the prelarvae was estimated from the caudal fin fragment using the CyStain DNA 2 step kit (Sysmex Europe GmbH, Norderstedt, Germany). A tissue sample of around 1–2 mm^−2^ was minced and incubated in 500 μL of the nucleus extraction buffer to segregate and permeabilize the cells. The cells were stained using a 2 mL staining solution, which contained the fluorescent DNA dye DAPI (4′, 6- diamidino-2-phenylindol). Thirty-five prelarvae samples from each group were processed for flow cytometry.

### 2.6. Statistical Analyses

Data analysis was performed with Statistica 13 (STATISTICA advanced, module STATISTICA Multivariate Exploratory Technique; Statsoft). The data were initially analyzed for normal distribution using the Kolmogorov–Smirnov test. Multiple comparisons of the fertilization rate, hatching rate, mean BW, specific growth rate and growth heterogeneity between different crosses were carried out by using the analysis of variance. Initially, the assumptions of ANOVA were checked, and when they were not respected, the Kruskal–Wallis test was performed. Later on, Tukey’s post hoc (for ANOVA) or Dunn’s post hoc (for Kruskal–Wallis test) tests were performed to evaluate the significant differences between the crosses. The differences in cumulative survival were determined using Pearson’s Chi-square test on the surface significance of α = 0.05. Statistical significance of the differences within and between the individual groups was tested by analysis of variance ANOVA with the significance of α = 0.05. The histograms for the relative DNA content in the parents and f1 generation were combined together by overlaying raw data in the *.fcs format using CYTO-SW 0.3 software (Wolf & Danniel s.r.o., Mlada Boleslav, Czech Republic).

## 3. Results

### 3.1. Confirmation of Ploidy Levels of Parents with Normal and Abnormal Ploidy

The results from the flow cytometry analysis of the broodstock revealed the relative DNA content of the three males of the *A. ruthenus* standard to be diploid (Figure 1, blue peak 2n), three males of *A. baerii* to be tetraploid (Figure 1, blue peak 4n) and one hexaploid specimen of *A. baerii* to be 3-fold of the diploid *A. ruthenus* (Figure 1, blue peak 6n). Likewise, the sperm of the analyzed males had an average relative DNA content equivalent to haploidy (Figure 1, red peak 1n), diploidy (Figure 1, red peak 2n) and triploidy (Figure 1, red peak 3n) in diploid *A. ruthenus,* tetraploid *A. baerii* and hexaploid *A. baerii,* respectively. Likewise, one pentaploid male of *A. baerii* (Figure 2, blue peak 5n) was around 2.5-fold that of the reference diploid sterlet (Figure 2, red peak 2n). The sperm of the analyzed pentaploid *A. baerii* (Figure 2, blue peak ~2.5n) had a 2.5-fold higher average relative DNA content than the *A. ruthenus* sperm (Figure 2, red peak 1n). The coefficient of variation (C_V_) in both the sperm and erythrocyte relative DNA content was equal to or lower than 3% for all specimens. 

### 3.2. Detection of Ploidy Levels in the F1 Generation Hybrids

The relative DNA contents of the reference diploid *A. ruthenus* and tetraploid *A. baerii* purebreds were 50.89 ± 2.14 and 101.56 ± 4.50, respectively (Table 2). The ploidy allotment of all the abnormal ploidy was done by comparing the relative DNA content with the reference diploid *A. ruthenus* and tetraploid *A. baerii* purebreds. The ploidy level of the purebred *A. ruthenus* and *A. baerii* were 100% diploid (Table 2; Figure 3, red and blue peak 2n) and tetraploid (Table 2; Figure 3, green peak 4n), respectively, whereas the reciprocal hybrids produced from *A. ruthenus* and normal *A. baerii* (S × St and St × S) were 100% triploids (Table 2; Figure 3, green peak 3n). The ploidy level of the hybrid between diploid *A. ruthenus* and pentaploid *A. baerii* (St × S 5n) was ~3.5n (Table 2; Figure 3, red peak ~3.5n), whereas the hybrids of diploid *A. ruthenus* and hexaploid *A. baerii* (St × S 6n) were 4n (Table 2; Figure 3, green peak 4n). Likewise, the ploidy level of the hybrid between tetraploid *A. baerii* and pentaploid *A. baerii* (S × S 5n) was ~4.5n (Table 2; Figure 3, blue peak ~4.5n), whereas the hybrids of tetraploid *A. baerii* and hexaploid *A. baerii* (S × S 6n) were 5n (Table 2).

### 3.3. Fertilization and Hatching

The highest fertilization rate (76.54 ± 2.3) was recorded in the St × S hybrid, which was not significantly different from the sterlet purebred, Siberian sturgeon purebred, St × S 5n hybrid and S × S 5n hybrid. The lowest fertilization rate (11.16 ± 0.74) was observed in the St × S 6n hybrid, which was significantly different than the other crosses (Table 3). The highest hatching rate (67.93 ± 2.60) was observed in the Siberian sturgeon purebred, which was not significantly different from the majority of the crosses except for St × S 6n, S × S 6n and S × S 5n. The lowest hatching rate (6.57 ± 2.35) was observed in the St × S 6n thawed hybrid, which was significantly different than the other crosses (Table 3).

### 3.4. Mean Body Weight and Cumulative Survival

The swim-up larvae of the St × S 5n hybrid showed 100% mortality after five days of hatching. Likewise, the hatched individuals of the St × S 6n hybrid showed 100% mortality after the first sampling point (22 dph). The observed mean BW and cumulative survival of the crosses during all the sampling periods are shown in Figure 4 and Figure 5. The highest mean BW (277.79 ± 127.79) was recorded in the St × S hybrid on 497 dph (Figure 4). The mean BW in the St × S hybrid was not significantly different from the S × S 5n hybrid on 497 dph. The recorded values of the mean BW in the S × S 5n hybrid was found to be significantly higher than the other groups on three sampling points (160, 252 and 330 dph). The highest cumulative survival (14.47 ± 5.70) was observed in the S × St hybrid on 497 dph (Figure 5). The S × St hybrid displayed a significantly higher cumulative survival than the other crosses on all the sampling points. The significant differences between the groups in the cumulative survival were not estimated for the first three sampling points since the groups were separately reared in different tanks till 160 dph; they were also not reared in replicate tanks. The sterlet purebred and S × S 5n hybrid displayed comparatively lower cumulative survival than other crosses at most sampling points.

### 3.5. Mid-Parental Heterosis, Specific Growth Rate and Growth Heterogeneity 

The mid-parental heterosis recorded in the S × St hybrid for cumulative survival was found to be the highest and positive for all the sampling points, whereas the S × S 5n hybrid displayed negative heterosis for cumulative survival at all the sampling time points (Figure 6). The observed values of SGR differed significantly among groups, from 22–65 dph, with the sterlet purebred (8.21 ± 0.68) exhibiting the highest SGR and S × S 6n hybrid (6.58 ± 0.41) the lowest. There were no significant between-group differences in SGR at later sampling points. The observed values of growth heterogeneity differed significantly among groups from 22–65 dph, with the S × S 5n hybrid exhibiting the highest GH and the S × S 6n hybrid the lowest. However, no significant GH differences were recorded between the groups at the later sampling points (Table 4).

## 4. Discussion 

It is well known in the literature that sturgeons are highly prone for hybridization as well as for spontaneous autopolyploidization [7,9,10,13]. Although the most probable hybrid origin of polyploid sturgeon species has been confirmed by various data [24], the bases of their evolutionary success of abnormal ploidy levels and exceptional genome plasticity are still poorly understood. Additionally, the scope for expansion of sturgeon aquaculture has constantly been increasing these days due to the high commercial importance of black caviar. Notably, a considerable amount of aquaculture caviar production directly comes from sturgeon hybrids [31]. This led to increased production of various sturgeon hybrids formed between two species that originated from different parental ploidy levels in commercial aquaculture farms for better vigor of the hybrids [18,28,38].

In this study, by using flow cytometry analysis, we initially confirmed the presence of the abnormal ploidy levels of sturgeon broodstock in Genetic Fisheries Center, Vodňany. The results from the flow cytometry analysis revealed that the erythrocyte relative DNA content of the *A. ruthenus* standards were diploid (2n), *A. baerii* were tetraploid (4n) and one suspected *A. baerii* male with abnormal ploidy was pentaploid (5n). The origin of the hexaploid and pentaploid *A. baerii* is probably due to a spontaneous polyploidy phenomenon. Spontaneous polyploidy already has been reported in several cultured fish species [13]. The presence of an additional set of chromosomes in diploid (2n) species gives rise to triploid (3n) individuals that are infertile or sub-sterile. In case of tetraploid (4n) sturgeon species, spontaneous polyploidization gives rise to fertile hexaploid (6n) individuals that are already reported in *A. baerii* [22], *A. mikadoi* [39] and *A. transmontanus* [23]. Ploidy levels of the sterlet and Siberian sturgeon broodstock used in the experiment corresponded to 2n and 4n as previously reported for these species [13,22]. Likewise, some scientists discussed the possible fertility of sturgeon hybrids having an intermediate DNA content referring to recent triploids (3n) or evolutionary hexaploids (6n) [19]. Using flow cytometry, these authors detected evolutionary pentaploid (5n) and evolutionary heptaploid (7n) individuals among sturgeon aquaculture stocks, and suggested that these individuals might have originated from hybridization of an evolutionary hexaploid specimen (6n) with an evolutionary tetraploid specimen (4n), and an evolutionary hexaploid specimen (6n) with an evolutionary octoploid specimen (8n), respectively.

We then produced the hybrids using the pentaploid *A. baerii* male with *A. ruthenus* and tetraploid *A. baerii* females. As mentioned before, the cryopreserved sperm sample of one hexaploid (6n) *A. baerii* confirmed with flow cytometry analysis from the previous year was also used for the production of hybrids with abnormal ploidy levels. The cryopreserved sperm sample of the hexaploid (6n ~368 chromosomes) *A. baerii* male that we used in our experiment was confirmed to be of spontaneous polyploid origin [13]. Fertilization of ova of *A. ruthenus* and *A. baerii* with normal ploidy with the sperm of pentaploid and hexaploid *A. baerii* produced fully viable progeny whose ploidy levels were found intermediate to those of the parents. We did not observe any evident rise in the percent of malformed non-viable larvae in any specific hybrid cross. Finally, we can conclude that the ploidy levels of the studied F1 generation hybrids were in accordance with the widely accepted assumption that the hybrids obtained from the breeding between parents with different ploidy levels are considered to have ploidy level intermediate to those of parental individuals [40].

Coming to the fitness-related traits, we observed a composite pattern for both heterosis and outbreeding depression in interspecific hybrids of sturgeons with abnormal ploidy levels. The swim-up larvae of St × S 5n hybrid did not survive after hatching, and the St × S 6n hybrid showed 100% mortality by reaching 65 dph. This might be due to extreme genetic incompatibility between their genomes and chromosome numbers. However, other hybrids, like the S × S 5n and St × S hybrids, displayed a significantly higher mean BW in most sampling points. Interestingly, the S × S 5n hybrid, which grew well, showed the lowest survival in the majority of the sampling points. The observed heterosis levels in the F1 generation hybrids changed depending on the trait or the time period considered in the present study. However, these levels were quite comparable with other studies dealing with heterosis in sturgeons [29,30]. The perspectives on the influence of hybridization on evolution have changed from time to time. Some authors suggested that hybrids are the “raw materials” of evolution and a creative source of functional adoption [41,42,43], whereas a few studies concluded hybridization as an evolutionary dead-end of a species [44]. Overall, in this study, the hybrids displayed better cumulative survival and mean BW in all the sampling points. However, the incidence of females with abnormal ploidy levels have not to be reported yet. It would be interesting to study the gamete interaction between sturgeon females and males with normal and abnormal ploidy levels. The differences in the growth rate, reproductive features, survivability, disease resistance and metabolic processes in polyploid sturgeons are the key considerations from a scientific and an applied point of view.

The significant effect of polyploidization on wild sturgeon populations has not been intensively investigated. Spontaneous polyploidization has been reported in wild populations of fishes like cobitids and cyprinids [45,46,47,48]. There is no reported evidence on the incidence of spontaneous polyploidization in the natural waters for sturgeons as far [13]. However, currently, climate change and global warming are the two biggest hurdles in the natural habitat that can expose sturgeon gametes to rapidly fluctuating temperature stresses that might affect the meiotic or fertilization disorders, resulting in an increased rate of autopolyploidization. Additionally, spontaneous polyploid individuals are already reported in farmed conditions and can be assigned to logistic delays during artificial propagation [48]. Accidental escapees from hatchery conditions to the wild environment can pose a potential threat to genetic diversity and disturb the ploidy level of the native species. Presently, restocking is the most widely applied management strategy to enhance declining fish populations affected by environmental degradation or overharvesting with considerable interest. Meanwhile, the reintroduction of *A. transmontanus* with an abnormal ploidy level into the wild waters, resulting in delayed reproductive maturation in the F2 generation and subsequent generations, are already reported in North America [49]. Such reintroduction programs without a proper genetic screening can easily contaminate the gene pool of the native stocks. As reported in the captive populations of sturgeons in the aquaculture farms, the fertile spontaneous polyploid individuals might spawn in natural waters, and they can result in progeny with reduced fertility. In addition to that, due to the growth properties, hybrids could be more prominent in food competition and can have access to limited food resources easily in the wild. This may generate F1 generation individuals with reduced fitness, and thus can have a harmful effect on the original net population genetic site and on the productivity and survival of the overall native population. Yet, despite the recent progress in the field of molecular genetics, we still have a lot to discover when it comes to the molecular mechanisms that are involved in chromosome duplication and elimination in early germ cells in cell cycle control in polyploid sturgeons, which need to be addressed in future studies.

## 5. Conclusions

This is the first study that describes the comparative performance evaluation of the F1 generation hybrids with the purebreds produced from normal and abnormal ploidy levels of Siberian sturgeon and sterlet. Most of the produced hybrids displayed a better growth rate and survivability compared to the purebreds, except the St × S 5n and St × S 6n hybrids. With the given importance of polyploid sturgeons in aquaculture, further studies on their performance in terms of reproduction, growth, stress, immune responses and metabolic processes should be studied, to expand our knowledge on the physiological and immunological differences in sturgeon hybrids of abnormal ploidy levels. 

## Figures and Tables

**Figure 1 genes-11-01164-f001:**
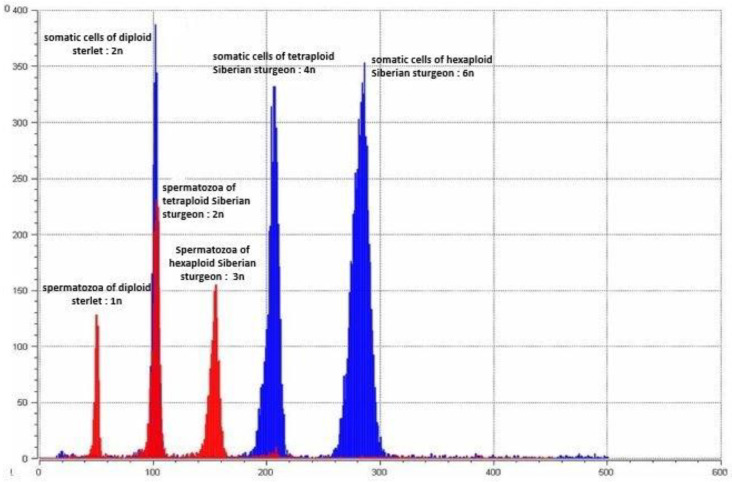
Flow cytometric histogram of the relative DNA content in somatic cells of parental fish (blue peaks) and corresponding spermatozoa (red peaks). Control diploid sterlet (*A. ruthenus*; peak 1n and 2n), tetraploid Siberian sturgeon (*A. baerii*; peaks 2n and 4n) and hexaploid Siberian sturgeon (peaks 3n and 6n).

**Figure 2 genes-11-01164-f002:**
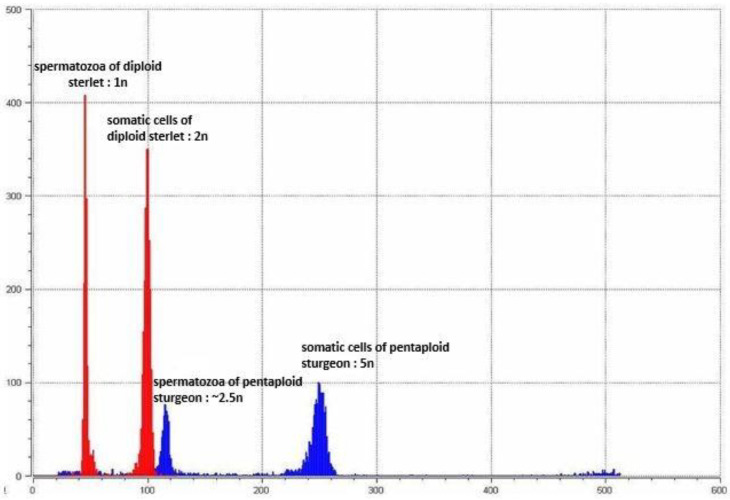
Flow cytometric histogram of relative DNA content in pentaploid Siberian sturgeon (*A. baerii*; blue peak and 5n) with spermatozoa (blue peak ~2.5n). Control diploid sterlet and sterlet spermatozoa (*A. ruthenus*; red peaks 1n and 2n) were taken as the reference ploidy.

**Figure 3 genes-11-01164-f003:**
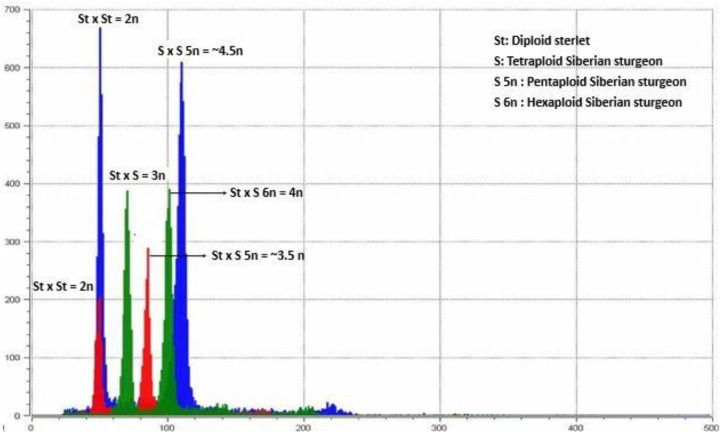
Flow cytometric histogram of the relative DNA content in the prelarvae of the control diploid (red and blue peaks 2n) sterlet purebred (*A. ruthenus*); the triploid (green peak 3n) hybrid of the diploid sterlet × tetraploid Siberian sturgeon (*A. baerii*); the hybrid with an intermediate ploidy level (red peak ~3.5n) of the diploid sterlet × pentaploid Siberian sturgeon; the tetraploid (green peak 4n) hybrid of the diploid sterlet × hexaploid Siberian sturgeon; and the hybrid with an intermediate ploidy level (blue peak ~4.5n) of the tetraploid Siberian sturgeon × pentaploid Siberian sturgeon.

**Figure 4 genes-11-01164-f004:**
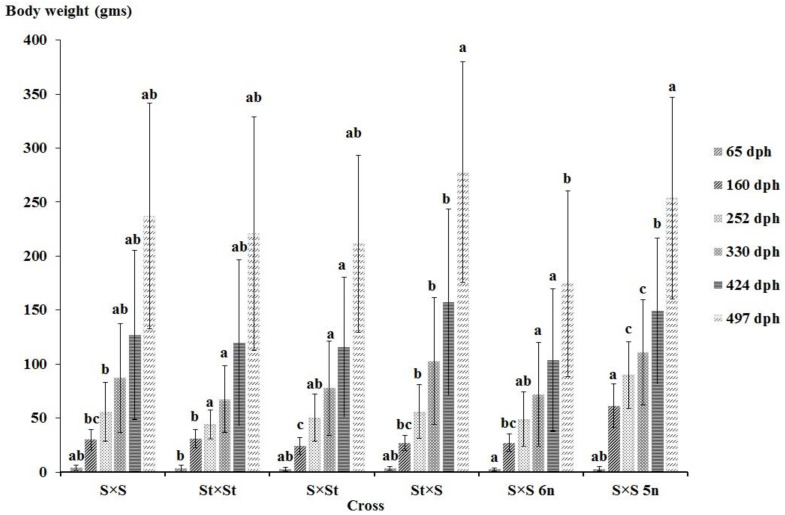
Mean bodyweight (g ± SD) of the produced crosses of different ploidy levels of Siberian sturgeon (*A. baerii*) and sterlet (*A. ruthenus*) at 65, 160, 252, 330, 424 and 497 dph. Columns with the same alphabetic superscript did not differ significantly at *p* < 0.05. St: diploid sterlet; S: tetraploid Siberian sturgeon; S 5n: pentaploid Siberian sturgeon; S 6n: hexaploid Siberian sturgeon.

**Figure 5 genes-11-01164-f005:**
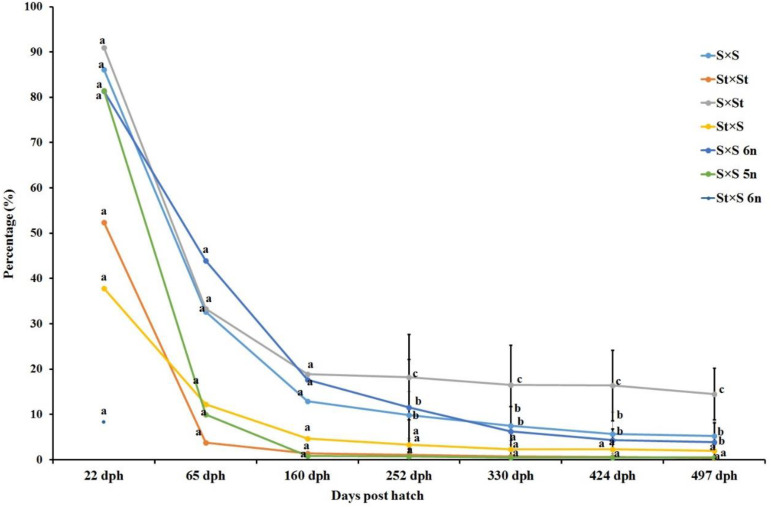
Cumulative survival (% ± SD) of the produced crosses of different ploidy levels of Siberian sturgeon (*A. baerii*) and sterlet (*A. ruthenus*) at 22, 65, 160, 252, 330, 424 and 497 dph. Marks with the same alphabetic superscript did not differ significantly at *p* < 0.05. St: diploid sterlet; S: tetraploid Siberian sturgeon; S 5n: pentaploid Siberian sturgeon; S 6n: hexaploid Siberian sturgeon.

**Figure 6 genes-11-01164-f006:**
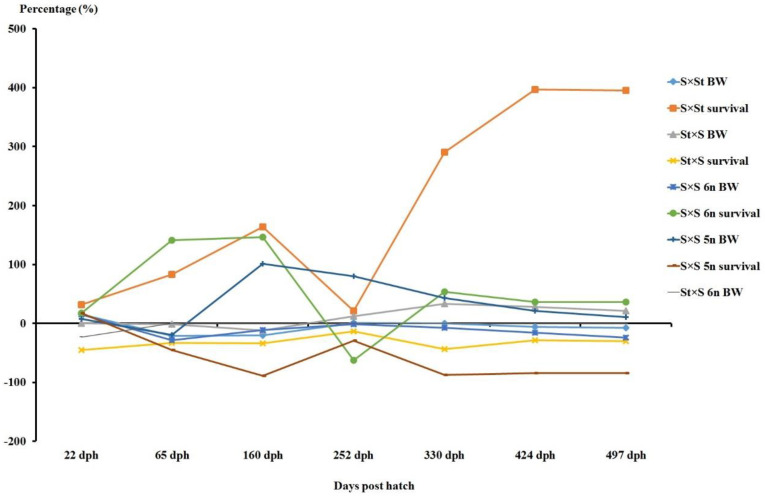
Estimated values of mid-parental heterosis (%) for mean bodyweight and cumulative survival among the produced crosses of different ploidy levels of Siberian sturgeon (*A. baerii*) and sterlet (*A. ruthenus*) at 22, 65, 160, 252, 330, 424 and 497 dph. St: diploid sterlet; S: tetraploid Siberian sturgeon; S 5n: pentaploid Siberian sturgeon; S 6n: hexaploid Siberian sturgeon; BW: mean bodyweight.

**Table 1 genes-11-01164-t001:** The relative volume of sperm kg^−1^ body mass (MB), concentration of spermatozoa (spz) and motility (mean ± SD) in sperm of sterlet (*A. ruthenus*), Siberian sturgeon (*A. baerii*), pentaploid Siberian sturgeon (*A. baerii*) and of the spontaneous hexaploid Siberian sturgeon (*A. baerii*). * denotes to multiplication.

Males	No. of Fish	Average Weight	Relative Volume of Sperm (mL kg^−1^)	Concentration (10^9^ mL^−1^)	Motility (% Motile spz)	Relative Total Sperm Count (* 10^9^ kg^−1^)
Sterlet	3	1.49 ± 0.2	21.63 ± 5.61	0.93 ± 0.21	28.33 ± 19.30	19.79 ± 6.64
Siberian sturgeon	3	5.8 ± 1.59	21.28 ± 8.44	0.15 ± 0.02	91.67 ± 2.35	3.10 ± 0.89
Pentaploid Siberian sturgeon	1	6.7 ± 0.00	34.33 ± 0.00	0.14 ± 0.00	55 ± 0.00	4.83 ± 0.00
Hexaploid Siberian sturgeon (frozen/thawed)	1	9.3 ± 0.00	24.29 ± 0.00	2.35 ± 0.00	15 ± 0.00	-

**Table 2 genes-11-01164-t002:** Ploidy levels of dams and sires of sterlet (*A. ruthenus*) and Siberian sturgeon (*A. baerii*) used in the hybridization matrix and of the offspring, assessed as the relative DNA content (mean ± S.D.) in somatic cells using flow cytometry. All ploidy levels are given on a functional scale.* All prelarvae in this group died during/after hatching.

Dam.		Sire		Offspring			
Species	Ploidy Level	Species	Ploidy Level	Relative DNA Content (Channel No.)	Coefficient of Variation (%)	Observed Ploidy Level	No. of Prelarvae Analyzed (N)
Sterlet	2n	Sterlet	2n	50.89 ± 2.14	2.10 ± 0.41	2n	33
Sterlet	2n	Siberian sturgeon	4n	73.50 ± 5.73	2.81 ± 0.27	3n	33
Sterlet	2n	Siberian sturgeon	5n	84.45 ± 2.36	3.14 ± 0.16	~3.5n	18 *
Sterlet	2n	Siberian sturgeon	6n	100.20 ± 5.21	2.64 ± 0.35	4n	33
Siberian sturgeon	4n	Siberian sturgeon	4n	101.56 ± 4.50	2.53 ± 0.32	4n	33
Siberian sturgeon	4n	Sterlet	2n	72.50 ± 8.42	2.63 ± 0.91	3n	33
Siberian sturgeon	4n	Siberian sturgeon	5n	116.54 ± 2.76	2.85 ± 0.13	~4.5n	33
Siberian sturgeon	4n	Siberian sturgeon	6n	124.87± 2.16	2.08 ± 0.47	5n	33

**Table 3 genes-11-01164-t003:** Fertilization and hatching rates among the produced hybrid and purebred crosses of different ploidy levels of Siberian sturgeon (*A. baerii*) and sterlet (*A. ruthenus*). Columns with the same alphabetic superscript did not differ significantly at *p* < 0.05. St: diploid sterlet; S: tetraploid Siberian sturgeon; S 5n: pentaploid Siberian sturgeon; S 6n: hexaploid Siberian sturgeon.

Cross	Fertilization Rate (% Mean ± SD)	Hatching Rate (% Mean ± SD)
St × St	64.6 ± 0.32 ^ab^	58.57 ± 1.33 ^ab^
St × S	76.54 ± 2.3 ^a^	58.72 ± 3.15 ^ab^
St × S 5n	67.49 ± 2.78 ^ab^	56.82 ± 5.24 ^ab^
St × S 6n	11.16 ± 0.74 ^c^	6.57 ± 2.35 ^d^
S × S 6n	28.89 ± 4.44 ^d^	18.80 ± 8.19 ^c^
S × S 5n	74.64 ± 0.36 ^ab^	50.75 ± 4.78 ^b^
S × St	67.54 ± 3.9 ^b^	59.68 ± 6.24 ^ab^
S × S	76.21 ± 1.21 ^a^	67.93 ± 2.60 ^a^

**Table 4 genes-11-01164-t004:** Specific growth rate and growth heterogeneity (mean ± SD) of the produced hybrid and purebred crosses of different ploidy levels of Siberian sturgeon (*A. baerii*) and sterlet (*A. ruthenus*) at 22, 65, 160, 252, 330, 424 and 497 dph. Different alphabetic superscripts within a column indicate significant difference at *p* < 0.05. St: diploid sterlet; S: tetraploid Siberian sturgeon; S 5n: pentaploid Siberian sturgeon; S 6n: hexaploid Siberian sturgeon.

Days Post Hatching	22–65 dph	65–160 dph	160–252 dph	252–330 dph	330–424 dph	424–497 dph
**Specific growth rate (% day^−1^)**						
S × S	7.38 ± 0.29 ^ab^	2.16 ± 0.08 ^a^	0.68 ± 0.12 ^a^	0.58 ± 0.34 ^a^	0.38 ± 0.11 ^a^	0.85 ± 0.22 ^a^
St × St	8.21 ± 0.68 ^b^	2.29 ± 0.46 ^a^	0.37 ± 0.04 ^a^	0.56 ± 0.12 ^a^	0.59 ± 0.11 ^a^	0.82 ± 0.62 ^a^
S × St	7.11 ± 0.33 ^ab^	2.05 ± 0.38 ^a^	0.48 ± 0.17 ^a^	0.62 ± 0.25 ^a^	0.41 ± 0.13 ^a^	0.83 ± 0.62 ^a^
St × S	7.51 ± 0.54 ^ab^	2.21 ± 0.23 ^a^	0.79 ± 0.01 ^a^	0.76 ± 0.41 ^a^	0.47 ± 0.29 ^a^	0.75 ± 0.12 ^a^
S × S 6n	6.58 ± 0.41 ^a^	2.52 ± 0.14 ^a^	0.59 ± 0.15 ^a^	0.49 ± 0.43 ^a^	0.37 ± 0.15 ^a^	0.71 ± 0.44 ^a^
S × S 5n	6.88 ± 0.68 ^ab^	3.28 ± 0.23 ^a^	0.42 ± 0.07 ^a^	0.26 ± 0.27 ^a^	0.33 ± 0.80 ^a^	0.71 ± 0.07 ^a^
St × S 6n	-	-	-	-	-	-
St × S 5n	-	-	-	-	-	-
**Growth heterogeneity**						
S × S	3.31 ± 0.14 ^a^	0.52 ± 0.04 ^a^	1.54 ± 0.16 ^a^	1.20 ± 0.38 ^a^	0.91 ± 0.33 ^a^	1.47 ± 0.48 ^a^
St × St	2.84 ± 0.81 ^ab^	0.37 ± 0.14 ^a^	1.18 ± 0.04 ^a^	1.44 ± 0.15 ^a^	1.27 ± 0.33 ^a^	0.94 ± 0.06 ^a^
S × St	2.95 ± 0.41 ^ab^	0.47 ± 0.03 ^a^	1.72 ± 0.90 ^a^	1.18 ± 0.37 ^a^	0.94 ± 0.13 ^a^	1.01 ± 0.11 ^a^
St × S	1.74 ± 0.40 ^b^	0.52 ± 0.11 ^a^	1.71 ± 0.05 ^a^	1.45 ± 0.62 ^a^	0.98 ± 0.10 ^a^	0.71 ± 0.05 ^a^
S × S 6n	1.65 ± 0.36 ^b^	0.52 ± 0.07 ^a^	1.83 ± 0.25 ^a^	1.15 ± 0.24 ^a^	0.99 ± 0.16 ^a^	0.83 ± 0.20 ^a^
S × S 5n	3.84 ± 1.08 ^a^	0.43 ± 0.09 ^a^	1.03 ± 0.06 ^a^	1.27 ± 0.29 ^a^	0.98 ± 0.07 ^a^	0.88 ± 0.02 ^a^
St × S 6n	-	-	-	-	-	-
St × S 5n	-	-	-	-	-	-

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
