# Peer review of "Ploidy Levels and Fitness-Related Traits in Purebreds and Hybrids Originating from Sterlet (Acipenser ruthenus) and Unusual Ploidy Levels of Siberian Sturgeon (A. baerii)"

_genes, 2020, doi:10.3390/genes11101164_

Round 1

Reviewer 1 Report

Line 33-39
Please provide more information about the sturgeon species. Such as distribution, classification (e.g. the genus in this study), etc

Line 56-65
Although these cases are generally described, it is still necessary to add what species it is.

Line 91
Show the complete genus name

Table 4, Table 5, it is not easy to understand, it is good to make as figures?

Could you show the figures as: Shivaramu, S.; Vuong, D.T.; Havelka, M.; Lebeda, I.; Kašpar, V.; Flajšhans, M. The heterosis estimates for growth and survival traits in sterlet and Siberian sturgeon purebreds and hybrids. J. Appl. Ichthyol. 2020, 36, 267–274.

Could you provide photos of your fish (5n, 6n?), is it show any differences in morphology?

Please check everywhere (especially in tables), show us like: sterlet (St; A. ruthenus) and Siberian sturgeon (S; A. baerii). I can not easy to understand which is S, St, Siberian sturgeon, A. baerii, sterlet A. ruthenus from Tables and Figures directly.

Why you have 8 mating groups (St x St; St x S; St x S5n; St x S6n; S x S6n; S x S5n; S x St; S x S), but do not show all the groups in table 4 and table 5?

Fig 1-3, Could you show the mating groups or some symbols (e.g. spermatozoa or spceis) on the figures directly, not only in Figure captions.

Author Response

Dear Reviewer,

Thank you so much for your valuable comments and suggestions. We have prepared the revised version of the manuscript addressing all your comments and doing necessary amendments. We hope you will be satisfied with this version.

Line 33-39
Please provide more information about the sturgeon species. Such as distribution, classification (e.g. the genus in this study), etc

Thanks a lot for this comment. The text has been added as suggested. Please refer to the text.

Line 56-65
Although these cases are generally described, it is still necessary to add what species it is.

We have altered the text according to your suggestion. Please refer to the text.

Line 91
Show the complete genus name

Done as suggested. Please refer to the text.

Table 4, Table 5, it is not easy to understand, it is good to make as figures?

Done as suggested. Please refer to the text.

Could you show the figures as: Shivaramu, S.; Vuong, D.T.; Havelka, M.; Lebeda, I.; Kašpar, V.; Flajšhans, M. The heterosis estimates for growth and survival traits in sterlet and Siberian sturgeon purebreds and hybrids. J. Appl. Ichthyol. 2020, 36, 267–274.

Done as suggested for the table 4. Please refer to the text. We think there will too many graphs if we will generate the graphs for table 5. Also, there were no significant differences in the growth heterogeneity and specific growth rate for most of the sampling points. Hence, we believe in keeping the table 5 as it is.

Could you provide photos of your fish (5n, 6n?), is it show any differences in morphology?

There were no significant differences in the morphology of pentaploid and hexaploid hybrids in different developmental stages. So, we thought it would not be necessary to capture the photos of fishes. Our sincere apologies in this regard. We don’t have the fishes at our facilities right now.

Please check everywhere (especially in tables), show us like: sterlet (St; A. ruthenus) and Siberian sturgeon (S; A. baerii). I can not easy to understand which is S, St, Siberian sturgeon, A. baerii, sterlet A. ruthenus from Tables and Figures directly.

The text has been altered as suggested. Please refer to the text.

Why you have 8 mating groups (St x St; St x S; St x S5n; St x S6n; S x S6n; S x S5n; S x St; S x S), but do not show all the groups in table 4 and table 5?

As mentioned in the results, the swim-up larvae of St×S 6n hybrid and  St×S 5n hybrid showed 100% mortality by 22 days post hatch. But, the rows are added as per your suggestion. Please refer to the text.

Fig 1-3, Could you show the mating groups or some symbols (e.g. spermatozoa or species) on the figures directly, not only in Figure captions.

Thanks a lot for this comment. It was really useful. Done as suggested. Please refer to the text.

Reviewer 2 Report

The topic of sturgeon ploidy has been the subject of scientific work for many years. The use of new research methods allows us to get to know this research topic better. The authors present very interesting research on the effects of obtaining individuals of various ploidy, analyze survival and growth.
This research is very valuable for aquaculture.
Results
For a clearer presentation of the results of the growth of hybrids of different ploidy can be shown in the graph.
Discussion
The authors write exhaustively about the potential benefits and negative effects of hybridization. They compare the obtained results with the available literature and also write about the dangers that may result from the migration of hybrids to the natural environment. The authors point to the need to continue research taking into account metabolic processes, physiology and immunology.

Author Response

Dear Reviewer,

Thank you so much for your valuable comments and suggestions. We have prepared the revised version of the manuscript addressing all your comments and doing necessary amendments. We hope you will be satisfied with this version.

The topic of sturgeon ploidy has been the subject of scientific work for many years. The use of new research methods allows us to get to know this research topic better. The authors present very interesting research on the effects of obtaining individuals of various ploidy, analyze survival and growth.
This research is very valuable for aquaculture.

Thank you so much for your positive comments on the manuscript. We have tried to improvise the manuscript during this revision. We hope you’ll be satisfied with the revised version.

Results
For a clearer presentation of the results of the growth of hybrids of different ploidy can be shown in the graph.

Thanks a lot for this comment. Done as suggested. Please refer to the text.

Discussion
The authors write exhaustively about the potential benefits and negative effects of hybridization. They compare the obtained results with the available literature and also write about the dangers that may result from the migration of hybrids to the natural environment. The authors point to the need to continue research taking into account metabolic processes, physiology and immunology.

Thanks a lot for the positive remarks. We hope you’ll be satisfied with this revised version.